# Biomineralized Nano-Assemblies of Poly(Ethylene Glycol) Derivative with Lanthanide Ions as Ratiometric Fluorescence Sensors for Detection of Water and Fe^3+^ Ions

**DOI:** 10.3390/polym14101997

**Published:** 2022-05-13

**Authors:** Tong Chen, Sanping Zhao

**Affiliations:** 1College of Materials Science and Engineering, Wuhan Textile University, Wuhan 430073, China; chentong1997518@163.com; 2State Key Laboratory of New Textile Materials and Advanced Processing Technologies, Wuhan Textile University, Wuhan 430073, China

**Keywords:** metal–organic frameworks, biomimetic mineralization, luminescent properties, fluorescent nanosensors

## Abstract

An effective strategy was developed to fabricate novel lanthanide ions–pyromellitic acid–methoxy poly(ethylene glycol) (Ln-PMA-MPEG) nano-assemblies. The amphiphilic partially esterified derivative (PMA-MPEG) of pyromellitic acid with methoxy poly(ethylene glycol) was designed and synthesized via the coupling reaction. Ln-PMA-MPEG nano-assemblies were rapidly fabricated using PMA-MPEG as a polymer ligand with Eu^3+^ ions or mixed Eu^3+^/Tb^3+^ ions through biomimetic mineralization in neutral aqueous systems. The size of the as-prepared materials could be designed in the range 80–200 nm with a uniform distribution. The materials were readily dispersed in various solvents and displayed visible color variations and different photoluminescent properties for solvent recognition. The mixed Eu/Tb-PMA-MPEG nanomaterials were investigated as ratiometric sensors for the detection of trace water in DMF and Fe^3+^ ions in aqueous solutions. The sensor materials can quantitatively detect trace water in DMF from 0% to 10% (*v*/*v*). The resultant materials also display a strong correlation between the double luminescence intensity ratios (*I*_Tb_/*I*_Eu_) and Fe^3+^ concentration, with a good linear detection concentration in the range of 0–0.24 mM and a limit of detection of 0.46 μM, and other metal ions did not interfere with the sensing mechanism for Fe^3+^ ions. The novel nano-assemblies have potential applications as ratiometric fluorescent nanosensors in the chemical industry as well as in biomedical fields.

## 1. Introduction

Fluorescence sensing has attracted great attention due to its convenience, sensitivity, specificity and fast response time [1]. Quantum dots [2], organic molecules [3], luminescent lanthanide (Ln) materials [4] and metal–organic framework materials (MOFs) [5] have often been employed for the construction of multifunctional fluorescent sensors. Among these, the unique optical characteristics of lanthanide ions (especially Eu^3+^ and Tb^3+^) are particularly intriguing for sensor design because of their narrow and easily identifiable emission bands, large Stokes shifts and long fluorescence lifetimes [6,7]. Ln^3+^ ions themselves display low fluorescence intensity resulting from their weak absorption efficiency caused by the forbidden nature of the f–f transition [8]. The coordination of Ln^3+^ ions with organic ligands with higher molar absorption can greatly enhance their absorption ability and greatly improve the quantum efficiency of Ln^3+^ ions due to the so-called “antenna effect” [9]. In contrast, MOFs, known as crystalline polymers generated from metal ions and organic ligands, have emerged as important candidates for chemosensory materials owing to their porous structures and high surface areas [10]. The self-assembly of Ln^3+^ ions with suitable organic linkers into porous MOFs can render a powerful luminescent platform [11]. Mixed-crystal LnMOFs, adopting two or more Ln ions, could be designed as ratiometric fluorescent sensors to afford more accurate measurement by virtue of the self-calibration of two signal peaks, avoiding the influence of several environmental factors on the absolute fluorescence intensity of a single emissive transition [12]. In recent years, a series of LnMOFs-based ratiometric sensor materials with specific sensing capabilities have been successfully prepared and used as sensors for the highly sensitive and selective detection of trace water in various solvents [13,14], small molecules [15,16], cations/anions [17,18], humidity [19], pH [20,21,22] and temperature [23]. However, most LnMOFs studied and applied are in the form of a crystalline powder, generally of submicron or micron size, which greatly limits their abilities to act as universally applicable sensors due to poor compatibility with other materials [24].

To date, LnMOFs are commonly prepared via hydrothermal or solvothermal approaches. High temperature, high pressure and a relatively long time are often necessary [13,25]. Recently, obtaining MOFs through biomimetic mineralization using biomacromolecules (peptide or protein) has attracted attention due to the economically and environmentally benign process, including neutral pH, room temperature, short time and no requirements for other chemicals. The typical MOFs such as ZIF-8, KHUST-1 and Eu/Tb BDC can be prepared via biomimetic mineralization process [26,27]. The biomimetic mineralization of Polymer@MOFs using a polymer as an organic ligand remains relatively understudied [28,29]. In previous work, we developed an effective strategy to fabricate nanoscale LnMOFs using a thermosensitive polymer bearing -COOH and -NH-CO- groups on the polymer chains as organic links with Ln ions through a biomimetic mineralization method [30]. Polymer@LnMOFs using polymer ligands could improve the compatibility of LnMOFs with other materials and the stability of LnMOFs, consequently greatly broadening their application areas [31,32].

Poly(ethylene glycol) (PEG), a synthetic polymer, has been widely used in biomedical applications due to its biocompatibility, hydrophilicity, non-immunogenicity and non-toxicity [33]. Pyromellitic acid (PMA), a kind of insoluble multicarboxylic acid aromatic molecule, is frequently investigated and employed as organic ligand to construct MOFs with metal ions [34]. Herein, the partially esterified derivative (PMA-MPEG) of PMA and MPEG (*M*_n_ = 1000) was designed and synthesized as a functional ligand that could form micelles with PMA moieties as the core and MPEG blocks as the corona in aqueous solution. In neutral aqueous medium, the luminescent Ln-PMA-MPEG nano-assemblies could be readily fabricated by the self-assembly of MPEG functionalized with PMA with Eu^3+^ ions or Eu^3+^/Tb^3+^ mixed ions via biomimetic mineralization process. The novel Eu-PMA-MPEG nano-assemblies readily dispersed in different solvents, and displayed variable emissive light colors visible to the naked eye and different fluorescence properties for quantitative identification. The Eu/Tb-PMA-MPEG nano-assemblies as ratiometric sensors were highly sensitive to trace water in DMF as well as to Fe^3+^ ions in aqueous systems. These luminescent nanomaterials have potential applications as multifunctional ratiometric nanosensors in the chemical industry and in biomedical fields.

## 2. Experimental

### 2.1. Materials and Measurements

Pyromellitic acid (PMA, 98%, Sigma, St. Louis, MO, USA). Methoxypoly(ethylene glycol) (MPEG, Sigma, St. Louis, MO, USA) of *M*_n_ = 1000 g/mol was used after drying under vacuum at 90 °C for 24 h. 4-(Dimethylamino) pyridine (DMAP, 99%) and 1,3-dicyclohexyl carbodiimide (DCC, 99%) were purchased from Aladdin Reagent Co., Ltd., Shanghai, China. Eu(NO_3_)_3_⋅6H_2_O, Tb(NO_3_)_3_⋅6H_2_O and bovine serum albumin (BSA) used for material preparation were reagent grade and used without further purification. Triethylamine and THF (Sinopharm Chemical Reagent Co., Ltd., Shanghai, China) were treated with CaH_2_ for four days and distilled just before use. All other chemicals were of analytical grade and used without further purification. The water used was Milli-Q ultrapure water (Millipore, Billerica, MA, USA).

### 2.2. Synthesis of Partially Esterified Derivative (PMA-MPEG)

To obtain the partially esterified derivative of PMA and MPEG, 2 g (7.87 mmol) of PMA was dissolved into 200 mL of dried THF under an argon atmosphere; 4.87 g (23.6 mmol) of DCC was introduced into the solution, the mixture was reacted at room temperature for 12 h, and the precipitate was removed by filtration. Then, 11.8 g (11.8 mmol) of MPEG, 1.44 g (11.8 mmol) of DMAP and 1.19 g (11.8 mmol) of triethylamine were added into the filtration solution and the mixture was continuously reacted under an argon atmosphere at room temperature for 48 h. The crude product was obtained using anhydrous ethyl ether. The product was dissolved in ultrapure water, stirred for 48 h, and the precipitate was then removed by centrifugation; finally, the partially esterified derivative was obtained by freeze-drying (yield: 89%).

### 2.3. Preparation of Eu-PMA-MPEG and Eu/Tb-PMA-MPEG

Ln-PMA-MPEG was prepared via a similar method for the biomimetically mineralized growth of MOFs [26]. Briefly, an aqueous solution (10 mL) containing PMA-MPEG derivative (0.5 g) and BSA (0.015 g) was prepared, and then mixed with a separate aqueous solution (5 mL) of Eu(NO_3_)_3_⋅6H_2_O or a mixture of Eu(NO_3_)_3_⋅6H_2_O and Tb(NO_3_)_3_⋅6H_2_O (Eu:Tb = 1:1) (0.9 g) at room temperature. The mixture quickly became slightly turbid. The mixture was subjected to dialysis (MWCO = 1000 Da) against distilled water for two days, and Ln-PMA-MPEG was obtained via freeze-drying approach (yield: 80%).

### 2.4. Characterization Techniques and Analysis

#### 2.4.1. ^1^H NMR Spectroscopy

^1^H NMR spectrum of the PMA-MPEG derivative was recorded at room temperature on a Bruker AVANCE NEO 600 at 600 MHz using CDCl_3_ as the solvent.

#### 2.4.2. Fourier Transform Infrared Spectroscopy

FT IR analysis of PMA, MPEG, PMA-MPEG and Ln-PMA-MPEG was conduced on a Bruker Tensor 27 using KBr discs at room temperature.

#### 2.4.3. Differential Scanning Calorimetry

DSC was employed to investigate the thermal behaviors of MPEG, PMA-MPEG_1.4_ and Ln-PMA-MPEG on a TA Instruments Q20 differential scanning calorimeter under a nitrogen atmosphere. The melting point and recrystallization measurements were performed at a heating and cooling rate of 10 °C min^−1^, respectively.

#### 2.4.4. Transmission Electron Microscopy

TEM was conducted on a HT7700 electron microscope (Hitachi, Tokyo, Japan) at an acceleration voltage of 100 kV. Dilute DMF solutions of the Eu-PMA-MPEG and Eu/Tb-PMA-MPEG (Eu/Tb = 1:1) were deposited in copper grids coated with carbon. Excess solvent was swept away by touching the edge of the grids with a small piece of filter paper, and the grids were allowed to dry at ambient temperature for 24 h before measurement.

#### 2.4.5. Particle Size Distribution

The particle sizes of Ln-PMA-MPEG were determined using a Zetasizer Nano ZS ZEN3600. Each formulation was analyzed in triplicate.

#### 2.4.6. X-ray Diffraction

Wide X-ray diffraction patterns of MPEG, PMA-MPEG derivative, Eu-PMA-MPEG were collected using a Panalytical X’Pert PRO X-ray diffractometer (PANalytical B. V., Almelo, The Netherlands) with Cu Kα (0.154 nm) radiation (40 kV, 40 mA) at a scanning speed of 2*θ* = 1°·min^−1^ over the range 2*θ* = 5–35°.

#### 2.4.7. Fluorescence Measurements

Fluorescence spectra were recorded with an F-2500 fluorescence spectrophotometer (Hitachi, Tokyo, Japan).

## 3. Results and Discussion

### 3.1. Synthesis and Micellar Behavior of Partially Esterified Derivative

MPEG (*M*_n_ = 1000), a semi-crystalline synthetic polymer, is soluble in water and many organic solvents. A water-soluble PMA-MPEG derivative was synthesized via the coupling reaction between the carboxyl groups of PMA and the terminal hydroxyl group of MPEG using DCC as the coupling agent and DMAP/Et_3_N as the catalyst (Figure 1), with the molar feed ratio of PMA to MPEG as 1:1.5. ^1^H NMR measurement was performed to obtain insight into the chemical structure of the derivative. Appendix A displays the ^1^H NMR spectrum of the derivative. The signal at 4.43 ppm belongs to the methylene protons of the MPEG-CH_2_C*H*_2_-OCO-PMA derivative, indicating the successful coupling reaction between MPEG and PMA [35]. The esterification degree of PMA could be determined from the integration area ratio of the C*H*_3_O- protons (3.39 ppm) in the MPEG chains and the protons (7.30 ppm) in the benzene ring, and was found to be 1.4 MPEG chains per PMA molecule, coded as PMA-MPEG_1.4_. The chemical composition calculated from ^1^H NMR analyses is nearly consistent with the feed composition.

The micelle formation by PMA-MPEG_1.4_ was verified by the fluorescence spectra using pyrene as the probe [36]. Figure 1a presents the emission spectra of pyrene in the presence of PMA-MPEG_1.4_. The intensity of the *I*_1_ peak gradually decreases with the incorporation of pyrene into the hydrophobic core region of the micelles from water, and the intensity ratio *I*_3_/*I*_1_ indicates the variation in micelle concentration. Figure 1b shows the intensity ratio of *I*_3_/*I*_1_ in the pyrene excitation spectra as a function of the logarithm of PMA-MPEG_1.4_ derivative concentration. The intersection of the two tangent curves, a horizontal curve at low derivative concentrations and the inflection, was determined to be the CMC; and the CMC value was 4.63 × 10^−4^ mg/mL. This value implied that the derivative had a very strong tendency to form micelles in aqueous solution.

### 3.2. The Structure and Morphology of Ln-PMA-MPEG

At present, many small-molecule carboxylic acids are frequently used as organic ligands for the preparation of LnMOFs because lanthanide ions exhibit a strong preference for negatively charged or neutral O donor atoms [37]. For the PMA-MPEG_1.4_ derivative, some unreacted -COOH residues remain in the PMA moieties. MPEG functionalized with PMA can be regarded as an organic ligand, and coordinates with lanthanide ions to form complexes due to the strong activity between lanthanide ions and O atoms from the -COOH groups of PMA (Figure 1). In addition, BSA, as a biological initiator, could accelerate the reaction process via biomimetic mineralization.

Figure 2 presents FT IR spectra of MPEG, PMA, PMA-MPEG_1.4_, Eu-PMA-MPEG and Eu/Tb-PMA-MPEG (Eu/Tb = 1:1). The absorption peaks between 3400 cm^−1^ and 3600 cm^−1^ are normally assigned to symmetric and asymmetric stretching vibration modes of the O-H bonds from -COOH; after coordinating with Eu^3+^ ions and Eu^3+^/Tb^3+^ mixed ions, the peaks become weaker and broader, and are shifted to a lower frequency (3400–3200 cm^−1^), which might be ascribed to the complexation between the lanthanide ions and the O atoms of O-H. The characteristic stretching peaks at 1730 cm^−1^ and 1640 cm^−1^ are the carbonyl (C = O) of the ester group and the -COOH groups, respectively; the bands become very weak in the IR spectra of Eu-PMA-MPEG and Eu/Tb-PMA-MPEG (Eu/Tb = 1:1). The lanthanide ions can coordinate with the two kinds of O from the ester group and the -COOH groups, which makes the carbonyl stretching vibration much weaker. The absorption peaks around 1000–1300 cm^−1^ are the -C-O- bands from the -COOH groups and ester groups, and the peaks became much weaker. After biomimetically mineralized growth of Ln-PMA-MPEG, the Ln ions could coordinate to the O atoms of the -COOH groups and the ester groups to form complexes with PMA moieties as organic ligands.

PEG is a typical semi-crystalline polymer. Here, MPEG with a low molecular weight (*M*_n_ = 1000) was used to improve the hydrophilicity of the insoluble PMA molecules and as-prepared nano-assemblies with Ln ions. DSC was employed to investigate the melting and crystallization behaviors of MPEG chains when introduced into PMA and Ln-PMA assemblies. Figure 3 depicts the second heating curves (a) and cooling curves (b) of MPEG, PMA-MPEG_1.4_ and Eu-PMA-MPEG. As seen from Figure 3a,b, the endothermal peak and the exothermal peak of the PMA-MPEG_1.4_ appeared at a relatively higher temperature (about 40 °C) and lower temperature (about 10 °C), respectively, compared with that of pure MPEG. The hydrogen bond interactions of unreacted carboxyl groups in PMA component were responsible for the acceleration of the crystallization process of MPEG moieties and the increased melting point of MPEG moieties. However, when the carboxyl groups of PMA-MPEG_1.4_ coordinated with Eu^3+^ ions, both the melting and crystallization behaviors of the MPEG component were destroyed, implying that the formation of Eu-PMA structure completely restricted the crystallization process of MPEG moieties to produce an amorphous state, which was consistent with previously reported results that the addition of nanoparticles caused a further reduction in the crystallinity of PEG [38]. It was also found that the Eu/Tb-PMA-MPEG sample was the same as Eu-PMA-MPEG, displaying a transparent and viscous fluid state at room temperature due to the amorphous structure of the MPEG component.

Further, the crystallinity of MPEG, PMA-MPEG_1.4_ and Eu-PMA-MPEG were investigated by XRD (Figure 4). As seen from Figure 4, pure MPEG displays the two main characteristic peaks at 19.1° and 23.2°, and the two characteristic peaks of MPEG moieties slightly become weaker after the reaction with PMA to produce PMA-MPEG_1.4_. However, for Eu-PMA-MPEG, neither crystal characteristics of MPEG moieties nor Eu-PMA assemblies could be observed, indicating that the semi-crystalline structure of MPEG moieties was completely disturbed after coordination with Eu^3+^ ions. The results are consistent with the DSC results (Figure 3a,b); the MPEG moieties in the Eu-PMA-MPEG exhibit an amorphous state without melting and crystallization behaviors. The viscous MPEG chains encapsulate the Eu-PMA nano-assemblies and may hinder the appearance of their crystalline characteristics.

Ln-PMA-MPEG samples dispersed easily in DMF, and the solutions are nearly clear. Size distributions of Eu-PMA-MPEG and Eu/Tb-PMA-MPEG (1:1) can be observed in Figure 5a,b, respectively. The particle sizes of Eu-PMA-MPEG and Eu/Tb-PMA-MPEG (1:1) were mainly in the range of 80–140 nm (91 nm 19%, 106 nm 46%, 122 nm 33%) and 100–200 nm (122 nm 23%, 142 nm 45%, 164 nm 28%), respectively. The nanoscale Ln-PMA-MPEG assemblies were fabricated successfully using PMA-MPEG_1.4_ as organic ligands with Ln^3+^ ions. TEM was further employed to examine the morphologies of the as-fabricated Ln-PMA-MPEG nano-assemblies. As shown in the insert Figure 5(a1,b1), the individual pristine Ln-PMA assemblies exhibited a nearly spherical morphology, and the particles of Eu-PMA assemblies were more uniform with a size of around 70 nm. The uniformity of the Eu/Tb-PMA assemblies was less than that of the Eu-PMA assemblies, and the particle size was about 90 nm, slightly larger than that of the Eu-PMA assemblies, which may be due to the different coordination ability between Eu^3+^ ions and Tb^3+^ ions with the neutral O atoms of the -COOH and ester groups from PMA moieties. The particle sizes of as-fabricated Ln-MPA-MPEG were relatively smaller and uniform, as compared with previously reported poly-LnMOFs prepared via biomimetic mineralization [30]. This may be ascribed to the micellar behaviors of PMA-MPEG_1.4_, which could act as nanoreactors regulating the growth of the Ln-PMA assemblies. The particle sizes examined from TEM were smaller than that in DMF solutions. This may be because TEM was used to examine the size of neat Ln-PMA assemblies, whereas the sizes measured in DMF were that of Ln-PMA particles anchored with MPEG chains, but the uniformity of the particles was coincided with that in DMF. The EDS results of PMA-MPEG_1.4_, Eu-PMA-MPEG and Eu/Tb-PMA-MPEG are shown in Appendix A, respectively. The content of Eu element in Eu-PMA-MPEG is 14.61 wt%, and Eu, Tb in the Eu/Tb-PMA-MPEG was 5.93 wt% and 9.75 wt%, respectively. The Pt element originates from the coating layer used for the test. Encapsulating MOFs in nanoshells has become one of the most promising strategies to overcome the stability issue of the MOFs. Besides, the activity and selectivity could be simultaneously enhanced by taking advantage of the synergy between the MOFs and the encapsulating materials, as well as the molecular sieving property of the encapsulating materials [39].

### 3.3. The Excitation Spectra and Emission Spectra of Ln-PMA-MPEG in Water and DMF

The excitation spectra of Eu-PMA-MPEG were investigated at 617 nm (^5^D_0_→^7^F_2_ transition of Eu^3+^) in water and DMF, respectively, as shown in Appendix A. There are multiple excitation peaks in the region of 260–450 nm; the peaks at 292 nm and 360 nm can be assigned to the absorption of PMA-MPEG ligands (black line in Appendix A). Excitation of the ligands at 360 nm results in a red line-shaped emission (red line in Appendix A). The emission bands are assigned to the transition from the ^5^D_0_ to ^7^F_J_ (J = 0–4) levels of Eu^3+^ at 578, 592, 617, 652 and 696 nm in DMF (Appendix A). However, the emission intensity becomes weak and two main emission peaks at 592 and 617 nm are observed in water due to the quenching of the water (Appendix A). The excitation spectra of Eu/Tb-PMA-MPEG were also evaluated at 544 nm (^5^D_4_→^7^F_5_ transition of Tb^3+^) and at 617 nm (^5^D_0_→^7^F_2_ transition of Eu^3+^) in water and DMF. In DMF (Appendix A), the transitions from the ^5^D_4_ to ^7^F_J_ (J = 6, 5 and 4) levels of Tb^3+^ at 490, 544 and 584 nm and the ^5^D_0_ to ^7^F_J_ (J = 1–4) levels of Eu^3+^ at 592, 617, 652 and 696 nm can be clearly observed. Despite the quenching of water, the two main emission peaks of Tb^3+^ (at 490 nm and 544 nm) and the two main emission peaks of Eu^3+^ (at 592 nm and 617 nm) are clearly present in the aqueous solution of Eu/Tb-PMA-MPEG. The above results illustrate the formation of Ln^3+^-PMA-MPEG complexes in the Ln-PMA-MPEG nano-assemblies and PMA-MPEG is a good sensitizer for Eu^3+^ and Tb^3+^ ions.

### 3.4. Fluorescent Properties of Eu-PMA-MPEG in Different Solvents for Solvent Recognition

Eu-PMA-MPEG was readily dispersed into different solvents. The fluorescent properties of Eu-PMA-MPEG were investigated in different solvents. Figure 6a presents the fluorescence spectra of Eu-PMA-MPEG at a concentration of 5 wt% in THF, H_2_O, acetone, ethanol and DMF investigated using a 360 nm excitation wavelength. As seen from Figure 6a, the peak at 390–480 nm is the emission of the ligand PMA-MPEG_1.4_. The peaks at 592 nm, 617 nm and 652 nm are the emission peaks of the Eu ions with the ^5^D_0_→^7^F_J_ (*J* = 1,2 and 3) transitions, respectively, and the peak at 617 nm is very intense, which is the typical red luminescence observed in Eu^3+^. Their fluorescence intensities change significantly in different solvents. The fluorescence intensity of Eu-PMA-MPEG is strongest in DMF, and then in THF, ethanol, acetone and water in sequence. There are several reasons for these variations. First, the solubility of Eu-PMA-MPEG in various solvents is different due to the various polarities (Figure 6c). Eu-PMA-MPEG is easily dispersed in DMF and the solution is nearly transparent; the aqueous solution of Eu-PMA-MPEG is slightly turbid; the solutions become increasingly turbid from THF, to ethanol and acetone, indicating that the Eu-PMA-MPEG particles have a tendency to aggregate in these solvents, and that the agglomeration of Eu-PMA-MPEG in these solvents could result in fluorescence quenching. At the same time, in these systems, energy transfer occurs between the PMA-MPEG_1.4_ ligand and Eu^3+^ ions, which is more complex than when small molecules are used as organic links; as seen from Figure 6a, the fluorescence intensities of PMA-MPEG_1.4_ obviously varied in different solvents. It is also possible that solvent molecules favor the formation of Eu-PMA complexes with large coordination numbers that can affect the energy transfer between PMA-MPEG_1.4_ and Eu^3+^ ions [40]. The abovementioned multiple effects lead to the fluorescence intensity variations in different solvents, which present different emission colors visible to the naked eye under 365 nm UV light (Figure 6d); the Eu-PMA-MPEG is bright blue in THF and bright red in DMF. The corresponding CIE diagram in different solvents is presented in the insert (Figure 6b). The observed emission colors are a good match to the calculated chromaticity coordinates.

### 3.5. Eu-PMA-MPEG as a Potential Fluorescent Sensor for Detection of Water in DMF

The detection of water and trace water in organic solvents is very important in many chemical industries because water is a common impurity affecting many chemical and industrial production processes [41]. As mentioned above, the fluorescence intensity of Eu-PMA-MPEG is very weak in water and strongest in DMF. The fluorescent properties of Eu-PMA-MPEG as a water sensor in water/DMF mixed solvents were investigated. Figure 7a exhibits the fluorescence spectra of Eu-PMA-MPEG in water/DMF mixed solvents with the water content varying from 0% to 100% (*v*/*v*). The broad peaks about 390–480 nm are the emission of the ligand PMA-MPEG_1.4_. The peaks at 592 nm and 617 nm are the emission peaks of Eu ions with the ^5^D_0_→^7^F_1_ transition and the ^5^D_0_→^7^F_2_ transition, respectively, and the peak at 617 nm is very intense in pure DMF. The fluorescence intensity *I*_Eu_ at 617 nm dramatically decreased when the water content reached only 10%. The *I*_Eu_ became more and more weak until there was nearly no fluorescence as the water content increased from 20% to 100%. From the optical photographs illuminated by 365 nm UV light (Figure 7b), variable colors can be clearly observed with naked eye when the water content increases from 0 to 20%.

As seen from Figure 7a, as the water content increases from 0 to 10% in DMF, the fluorescence intensity of Eu-PMA-MPEG is significantly quenched. Therefore, the fluorescence spectra of Eu-PMA-MPEG in trace water/DMF mixed solvents were studied as the water content increased from 0% to 10% (Figure 7c). As seen in Figure 7d, *I*_Eu_ at 617 nm gradually decreases with the increasing water content from 0% to 10%. The insert Figure 7(d1) shows the trend in *I*_Eu_ at 617 nm versus the water content could be well fitted to the second-order exponential equation y = 2112.1 × exp (−x/0.04) + 2127.7 × exp (−x/0.04) + 905.8 (R^2^ = 0.998). The *I*_0_/*I* ratio to water content follows the Stern–Volmer equation *I*_0_/*I* = 1 + K_SV_[H_2_O] (K_SV_ = 31.74) [42], where K_SV_ is the Stern–Volmer quenching constant, and *I*_0_ and *I* are the fluorescence intensities of Eu-PMA-MPEG at 617 nm in pure DMF and different water content, respectively. The Stern–Volmer plot in Figure 7e shows an adequate linear fit with a correlation coefficient (R^2^) of 0.998 between *I*_0_/*I* and the water content in the water content range of 0–10%. Therefore, Eu-PMA-MPEG as a trace water sensor in water/DMF solutions is more sensitive and exact.

### 3.6. Eu/Tb-PMA-MPEG as a Potential Ratiometric Fluorescent Sensor for Detection of Water in DMF

Most of the luminescent materials as trace detectors consisting of a single emission center are susceptible to many factors, from the detector concentration, excitation power, and optoelectronic system drift [43]. Eu/Tb mixed LnMOFs employed for detection could be self-calibrated. Herein, the fluorescent water sensing properties of Eu/Tb-PMA-MPEG are also evaluated. Figure 8a shows the fluorescence spectra of Eu/Tb-PMA-MPEG (Eu:Tb = 1:1). The fluorescence spectra of Eu/Tb-PMA-MPEG exhibit both the typical emission peaks of Eu ions (at 592 nm and 617 nm) and the typical emission peaks of Tb ions (at 490 nm and 544 nm). The fluorescence intensity *I*_Tb_ (at 544 nm, typical green luminescence of Tb ions) is much stronger than *I*_Eu_ (617 nm, typical red luminescence of Eu ions) due to the different coordination ability and excited state energy levels of Tb and Eu ions. The peaks at 544 nm and 617 nm in pure DMF are very strong; when the water content reaches 10%, *I*_Tb_/*I*_Eu_ dramatically decreases, and the reason is that the quenching efficiency of water to Eu^3+^ is higher than Tb^3+^ [13]. As the water content increases from 0 to 10%, the fluorescence intensity of Eu^3+^ quenches more significantly than Tb^3+^, which eventually leads to a substantial increase in *I*_Tb_/*I*_Eu_. The water content from 10% to 40%, *I*_Tb_/*I*_Eu_ gradually decreases. The change that occurs in response to DMF with a water content of 0%–10%–20% (*v*/*v*) is visible to the naked eye (Figure 8b), and the weak emission peak of Eu^3+^ at 652 nm and 696 nm disappeared when the water content reaches 20%. Then from 60% to 100%, *I*_Tb_/*I*_Eu_ had little change (Figure 8d). Further, the fluorescence spectra of the Eu/Tb-PMA-MPEG in DMF solutions containing trace water from 0% to 10% are investigated (Figure 8c), as seen from Figure 8c, *I*_Tb_ and *I*_Eu_ gradually decrease, the insert Figure 8e shows *I*_Tb_/*I*_Eu_ versus the water content is a good match for a linear relationship, and the linear equation is y = 2.39 + 17.69x (R^2^ = 0.997). A promising value as compared with the previously literature [44], where lanthanide-based solid-state sensors could achieve the quantitative detection over the range 10–120,000 ppm H_2_O in D_2_O. With the aid of dual-response luminescence centers, the mixed Eu/Tb-PMA-MPEG has promising potential application as ratiometric nanosensor in the field of trace water detection in the chemical industry.

### 3.7. Eu/Tb-PMA-MPEG as a Potential Ratiometric Fluorescent Sensor for Detection of Aqueous Fe^3+^ Ions

Fe^3+^ is an essential trace element in most physiological processes and it is also significant in water quality assessment. Therefore, the detection of Fe^3+^ is critical for the early identification and diagnosis of diseases caused by its excess or deficiency [45]. To evaluate the sensitivity and selectivity of Eu/Tb-PMA-MPEG to different metal ions, changes in fluorescence properties of Eu/Tb-PMA-MPEG upon addition of various metal ion salts or mixed metal ion salts are investigated (Figure 9a). The change in *I*_Tb_/*I*_Eu_ of Eu/Tb-PMA-MPEG in the absence or presence of metal ions or mixed metal ions is presented in Figure 9b, where *I*_Tb_ is the fluorescence intensity of Tb^3+^ ions at 544 nm and *I*_Eu_ is the fluorescence intensity of Eu^3+^ ions at 617 nm. Significant fluorescent quenching is observed for Eu/Tb-PMA-MPEG solution in the presence of Fe^3+^ ions with a concentration of 0.02 M under the excitation of 360 nm (Figure 9a), while the solutions with the same concentration of other metal ions result in small (in the case of Co^2+^ ion) or no obvious (in the case of K^+^, Al^3+^, Mg^2+^, Zn^2+^, Ba^2+^, Cu^2+^ ions) fluorescence changes. The potential interfering ions are also tested to check the selectivity of Eu/Tb-PMA-MPEG towards metal ions. Common mixed metal ions (including K^+^, Al^3+^, Mg^2+^, Zn^2+^, Ba^2+^, Cu^2+^ and Co^2+^) were tested at the same concentration of each ion as Fe^3+^ ions (0.02 M). The significant decay on photoluminescence intensity of mixed metal ions containing Fe^3+^ ions is observed under the excitation of 360 nm while there is slight increase in fluorescence intensity of mixed metal ions, as compared to that of Eu/Tb-PMA-MPEG aqueous solution (blank sample). These results show that the high sensitivity and selectivity of as-prepared Eu/Tb-PMA-MPEG toward Fe^3+^ ions compared to other metal ions, and there was no interference with other metal ions, indicating that Eu/Tb-PMA-MPEG is a stable luminescent Fe^3+^ sensor [46].

Further, for a sensitivity study, different concentrations of Fe^3+^ in the range of 0.002–2 mM were measured. Figure 9c shows the fluorescence spectra of the nanoscale Eu/Tb-PMA-MPEG solutions after adding various amounts of Fe^3+^ ions under the excitation wavelength of 360 nm. The fluorescence intensities *I*_PMA-MPEG_ (the very broad peak about 390–480 nm), *I*_Tb_ (the peak at 544 nm) and *I*_Eu_ (the peak at 617 nm) all decrease with the increase in Fe^3+^ concentration. Increasing Fe^3+^ concentration leads to an obvious drop in Tb^3+^ emission intensity, whereas the emission intensity of Eu^3+^ exhibits a slow decrease. This different Fe^3+^-dependent emission pattern of ^5^D_4_→^7^F_5_ transition (Tb^3+^, *I*_Tb_) and ^5^D_0_→^7^F_2_ transition (Eu^3+^, *I*_Eu_) in the Eu/Tb-PMA-MPEG can used as a promising candidate for self-referencing sensor. As seen from Figure 9d, the change rule between *I*_Tb_/*I*_Eu_ and the concentration of Fe^3+^ was clear. The plot in Figure 9e shows an adequate linear fit with a correlation coefficient (R^2^) of 0.998 between *I*_Tb_/*I*_Eu_ and the concentration of Fe^3+^ in the concentration range of 0–0.24 mM, and the linear equation is *I*_Tb_/*I*_Eu_ = 5.17–1.86C (Fe^3+^). The limit of detection (LOD) was found to be 0.46 μM, a lower value compared with that previously reported in the literature [47,48]. Appendix A presents the CIE chromaticity of the Eu/Tb-PMA-MPEG solutions after adding various amounts of Fe^3+^ ions under the excitation wavelength of 360 nm. The quenching mechanism could be explained as the interaction between Fe^3+^ and the PMA-MPEG_1.4_ ligand reduces the energy transfer efficiency from PMA-MPEG_1.4_ ligands to Eu^3+^ and Tb^3+^ ions, especially Tb^3+^ ions in Eu/Tb-PMA-MPEG, as seen in Figure 9c, and this effect increases with the addition of Fe^3+^; thus, the luminescence of the nanomaterial is quickly reduced.

In aqueous media, the characteristic emission intensity (*I*_Eu_) of Eu^3+^ ions at 617 nm in Eu-MPA-MPEG is very low due to the quenching caused from water (as shown in Appendix A), but for Eu/Tb-PMA-MPEG (Appendix A), the *I*_Tb_ of Tb^3+^ ions at 544 nm is much higher than thw *I*_Eu_ of Eu^3+^ ions at 617 nm, and *I*_Eu_ in Eu/Tb-PMA-MPEG is also higher than that in Eu-PMA-MPEG, which is ascribed to the energy transfer between lanthanides [49]. Such a self-referring strategy amplifies the relative emission ratios, which would also enhance the luminescence signals and facilitate the detection of the analytes in aqueous systems. Using nanoscale Ln-assemblies, especially those easily dispersed in liquid, to achieve fluorescence detection is fast, simple and accurate [50,51]. The mixed Eu/Tb-PMA-MPEG nanomaterials have promising potential application as ratiometric fluorescence nanosensor in biomedical fields due to the biocompatibility and hydrophilicity of PEG.

## 4. Conclusions

In conclusion, a facile strategy to fabricate novel Ln-PMA-MPEG nano-assemblies using a pyromellitic acid-poly(ethylene glycol) derivative as an organic ligand through a biomimetic mineralization method. The particle size of the Ln-PMA-MPEG is around 80–200 nm. The Ln-PMA-MPEG nanomaterials are readily dispersed into various solvents. Eu-PMA-MPEG could rapidly identify different solvents and quantitatively detect the water content in DMF. The mixed Eu/Tb-PMA-MPEG was highly sensitive to trace water in DMF and to Fe^3+^ in aqueous solutions without interference with other metal ions. The *I*_Tb_ (544 nm)/*I*_Eu_ (617 nm) and trace water, as well as Fe^3+^, were a good fit to a linear equation well. Eu/Tb-PMA-MPEG can be employed as potential ratiometric water and Fe^3+^ sensors. These Ln-PMA-MPEG nanomaterials have potential applications as ratiometric fluorescence nanosensors in the field of chemical industry or biomedical fields due to the biocompatibility and hydrophilicity of PEG.

## Data Availability

Not applicable.

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
