# Peer review of "Biomineralized Nano-Assemblies of Poly(Ethylene Glycol) Derivative with Lanthanide Ions as Ratiometric Fluorescence Sensors for Detection of Water and Fe3+ Ions"

_polymers, 2022, doi:10.3390/polym14101997_

Round 1

Reviewer 1 Report

The Chen and Zhao manuscript is devoted to new hybrid materials based on complexes of europium and terbium with derivatives of melitic acid and their application as luminescent sensors.

The article is large and contains a lot of interesting data. After some corrections, it may be published in Polymers.

Reviewer's notes:

  • Notes related to terminology:
    1. The term "MOF" is usually applied to ordered systems having a crystalline structure and containing sites - metal cations - and linker ligands. In this work, the synthesis products are X-ray amorphous, the assumptions about the connection of substituted melitic acid anions belonging to different “molecules” of the polymer through lanthanide ions have not been unambiguously proven and cannot apparently be proven. In this regard, according to the reviewer, the term MOF or nano-MOF is not correct to apply to the studied objects!
    2. The term "hybrids" or "hybrid materials" is traditionally applied to complex composites, two components of which (inorganic and organic) are linked by a covalent bond. In this paper, the materials do not have an inorganic core (such as SiO2, TiO2 etc), so how appropriate is the term "hybrid"? Wouldn't it be better to just talk about materials or nanomaterials?
  • Introduction and Literature Notes

There are few references to the literature in the introduction (27). I can recommend inserting a section with links to articles devoted to ratiometric sensor materials based on lanthanide bimetallic complexes. Sensors for H2O   (10.1016/j.chempr.2017.02.010 10.1109/JSEN.2019.2916498 10.1039/c9cc02324k), cations (such as  K+ 10.1039/c7tc04580h ), ammines  10.1039/c5tc04377h, Bioorganic analytes (10.1016/j.snb.2017.10.180 ) etc.  It will be useful to compare the obtained data on the sensitivity and range of the measured water concentration with analogues from the literature. In addition, examples of the synthesis of polymers based on polyethylene glycol and lanthanide carboxylates can also be found in the literature.

  • Representation Considerations
    1. Scheme 1 (Page 4). Why is one of the bonds in the benzene ring highlighted in bold and not shown as a double? The font size of the carboxyl groups in the right picture is too small. The parenthesis next to the inscription "n-1" gets on the link, 0 is used to indicate the temperature instead of the degree icon, etc. - it is necessary to redraw this and other schemes neatly and clearly.
    2. Figure 1, Page 4: this spectrum can be left in SI. The structure of the molecule in the figure is a curve.
    3. Why are the emission spectra Fig 7-10 only shown up to 650 nm? Europium has a strong 5D0-7F4 transition at ~700 nm, so spectra up to ~720 nm should be shown.
    4. Electronic transitions in the luminescence spectra of Eu and EuTb compounds must be labeled.
    5. Optionally, color coordinates can be given for luminescent materials in the presence of analytes - water and iron cations, in order to visually show the change in luminescence color.
  • Other remarks and questions
    1. Page 4 line 155: correct multiplication symbol 4.63◊10-4 mg/mL
    2. Page 5 line 160 fix multiplication symbol
    3. Page 9 line 251: in addition to the 5D0-7F1 transition, an electronic transition of europium 5D0-7F0 should be observed near ~580 nm. If its intensity is too low or the line is too wide, this requires explanation. What do electronic transitions look like with j=3 and 4?
    4. Would it be desirable to bring the luminescence excitation spectra?
    5. What is the content of terbium and europium in the obtained materials? Has elemental analysis been performed?
    6. How resistant are the materials to washing out of lanthanide with water?

Author Response

请看附件

Reviewer 2 Report

In this work, an effective strategy was developed to fabricate novel nanoscale methoxy poly(ethylene glycol) (MPEG)-lanthanide metal-organic frameworks (MPEG-LnMOFs) hybrids. It is a well-presented study, but there are some major issues to be addressed:

1) The whole paper has 37 cited Refs, and in the Intriuction there are 27. So, authors should drastically increase this part by citing more references in Discussion section.

2) The discussion is poor (see comment#1). Authors should make comparisons with already published literature to strentghent this part.

3) 2.4.1. section is not correctly numbered (there is not 2.4 so it cannot be 2.4.1)

4) 2.4.1 - 2.10 sections are recommended to nbe merged into "2.4 Characterizations techniques and Analysis"

5) The Introduction is too short. It needs re-writing.

6) The real novelty of the study should be clearly highlighted in the Introduction section.

Round 2

Reviewer 1 Report

The manuscript has been significantly improved. With the use of the term MOF, I cannot agree, but I will not insist. However, the literature review in the Introduction is very incomplete (27 references), I recommend adding more references to make the state of affairs in the field clear to the reader.

Li, H.; Han, W.; Lv, R.; Zhai, A.; Li, X. L.; Gu, W.; Liu, X. Dual-Function Mixed-Lanthanide Metal-Organic Framework for Ratiometric Water Detection in Bioethanol and Temperature Sensing. Anal. Chem. 2019, 91 (3), 2148–2154. https://doi.org/10.1021/acs.analchem.8b04690.

Gontcharenko, V. E.; Lunev, A. M.; Taydakov, I. V.; Korshunov, V. M.; Drozdov, A. A.; Belousov, Y. A. Luminescent Lanthanide-Based Sensor for H 2 O Detection in Aprotic Solvents and D 2 O. IEEE Sens. J. 2019, 19 (17), 7365–7372. https://doi.org/10.1109/JSEN.2019.2916498.

Xia, D.; Li, J.; Li, W.; Jiang, L.; Li, G. Lanthanides-Based Multifunctional Luminescent Films for Ratiometric Humidity Sensing, Information Storage, and Colored Coating. J. Lumin. 2021, 231. https://doi.org/10.1016/j.jlumin.2020.117784.

Li, H.; Liu, B.; Xu, L.; Jiao, H. A Hetero-MOF-Based Bifunctional Ratiometric Fluorescence Sensor for PH and Water Detection. Dalt. Trans. 2020. https://doi.org/10.1039/d0dt03626a.

Xia, T.; Zhu, F.; Jiang, K.; Cui, Y.; Yang, Y.; Qian, G. A Luminescent Ratiometric PH Sensor Based on a Nanoscale and Biocompatible Eu/Tb-Mixed MOF. Dalt. Trans. 2017, 46 (23), 7549–7555. https://doi.org/10.1039/c7dt01604b.

Zeng, X.; Hu, J.; Zhang, M.; Wang, F.; Wu, L.; Hou, X. Visual Detection of Fluoride Anions Using Mixed Lanthanide Metal-Organic Frameworks with a Smartphone. Anal. Chem. 2020, 92 (2), 2097–2102. https://doi.org/10.1021/acs.analchem.9b04598.

Ma, H.; Song, B.; Wang, Y.; Cong, D.; Jiang, Y.; Yuan, J. Dual-Emissive Nanoarchitecture of Lanthanide-Complex-Modified Silica Particles for in Vivo Ratiometric Time-Gated Luminescence Imaging of Hypochlorous Acid. Chem. Sci. 2016, 8 (1), 150–159. https://doi.org/10.1039/C6SC02243J.

Reviewer 2 Report

All my comments of the initial submission have been correctly replied and included in the revised manuscript. The quality of this work has been drastically improved after revision and therefore I recommend its publication as it is.

Author Response

Thank you for your reviewing our manuscript and recognizing our work!